

# Automated Detection and Chemical Characterization of Anomalous Grains in Scientific Ocean Drilling Legacy Cores

Gerald Auer[1], David De Vleeschouwer[2], Arisa Seki[3], Anna Joy Drury[4,5], Yusuke Kubo[6], Minoru Ikehara[7], Junichiro Kuroda[8], and the ReC23-01 scientists*

[1]Department of Earth Sciences, NAWI Graz Geocenter, University of Graz, Graz, 8010, Austria
[2]Institute of Geology and Palaeontology, University of Münster, Münster, Germany
[3]Faculty of Science, Shinshu University, Matsumoto, Nagano, Japan (now at Fukada Geological Institute, Bunkyo-ku, Tokyo, Japan)
[4]School of Geography, Geology, and the Environment, University of Leicester, Leicester, UK
[5]Department of Earth Sciences, University College London, London, UK
[6]Kochi Institute for Core Sample Research, Japan Agency for Marine-Earth Science and Technology, Kochi, Japan
[7]Marine Core Research Institute, Kochi University, Kochi, Japan
[8]Atmosphere and Ocean Research Institute, The University of Tokyo, Kashiwa, Chiba, Japan
*see team list for all team authors and their affiliations

*Correspondence to*: Gerald Auer (gerald.auer@uni-graz.at)

**Abstract.** The dynamic behavior of glaciers under varying climatic conditions plays a crucial role in Earth's climate systems, necessitating reliable records of glacial extent throughout the Cenozoic era. Ice-rafted debris (IRD) provides insights into past iceberg activity, glacial erosion rates, sediment transport, and meltwater delivery. Polar IRD reconstructions can enhance our understanding of ice sheet responses to climate shifts, however, traditional methods for detecting IRD are often destructive and labor-intensive. This study proposes a new methodological framework for detecting IRD in marine sediments as a proxy for reconstructing polar paleoclimate variability and glacial dynamics. Our new multi-proxy approach utilizes micro-X-ray fluorescence (µXRF) 2D imaging and computed tomography (CT) 3D scanning on five Deep Sea Drilling Project (DSDP) Site 266 archive half-sections. Our findings reveal that non-destructive scanning techniques have significant potential to effectively identify and quantify IRD, with µXRF providing high-resolution geochemical mapping and CT imaging enabling three-dimensional visualization of sediment structures. Machine learning classification of CT data using the Waikato Environment for Knowledge Analysis (WEKA) in the Fiji software package is a powerful tool for fast detection of anomalous components in IODP cores. However, we established that the finding of such coupled machine learning-CT approaches must be ground-truthed to ensure accuracy: our application of automated (k-means cluster analyses) chemical fingerprinting based on the µXRF data revealed many of the grains identified in the CT classification were false positives due to contamination from drill pipe flakes. We confirmed this contamination using scanning electron microscopy energy dispersive spectroscopy (SEM-EDS) on targeted samples. Fundamentally, our multi-proxy approach underscores that this approach alone cannot provide the geochemical fingerprinting necessary for unequivocal IRD identification. We therefore recommend a combined approach applying µXRF for geochemical fingerprinting and CT for structural detection to ensure a comprehensive, non-destructive method for IRD verification and preliminary provenance analysis. Ultimately, our work underscores the importance of applying stringent ground truthing to machine learning and high-resolution image datasets to ensure the data quality necessary for robust palaeoclimate reconstructions.




Keywords: Ice-rafted debris, Machine-learning in Geosciences, Non-destructive paleoclimate proxies, Marine sediment cores.

## 1 Introduction

The dynamic and geographical range of glaciers under variable climatic conditions is a key feedback in the Earth's climate
systems. Thus, generating reliable records for glacial extent throughout the Cenozoic is critical. Ice-rafted debris (IRD) in
marine sediments, in particular, is a crucial proxy to reconstruct polar paleoclimate variability and glacial dynamics. The
presence and distribution of IRD inform interpretations of past iceberg activity, glacial erosion rates, sediment transport, and
meltwater delivery (Grobe, 1987; McKay et al., 2022; Patterson et al., 2014; Starr et al., 2021; von Huene et al., 1973; Williams
et al., 2010). Generating reliable IRD records is thus important to understanding how ice sheets responded to climate shifts
over geological timescales.
Early methods developed for IRD detection and quantification focused on quantifying sand and gravel content over 20–30 cm
core sections (Conolly and Ewing, 1965; Smith et al., 1983) or sieving 100 g of dry sediment to measure pebble content within
the 1–2 cm size range (Vorren et al., 1989). Based on these early approaches, methodologies for detecting ice-rafted debris
(IRD) have evolved significantly over the past decades. Due to the limited sample availability in scientific drilling cores,
particularly in polar regions, less destructive methods gained traction. These approaches primarily rely on manual picking and
counting IRD from sieved sediment residues within defined size fractions, and were standard practice until the late 1980s
(Grobe, 1987). Such methods include manually analysing angular sand-sized quartz grains (63–250 µm) under a microscope
(Ledbetter and Watkins, 1978; Watkins et al., 1974, 1982) or counting lithic components in the >63 µm fraction of sediment
by weight (Labeyrie et al., 1986). Generally, grain size-based IRD quantification, including weight percentages of specific
fractions (e.g., >63 µm or 63-1000 µm), has been the most commonly used method (Bornhold, 1983; Cooke and Hays, 1982;
Kent et al., 1971; Piper and Brisco, 1975). However, this approach requires corrections for biogenic, volcanic, or diagenetic
grains, adding complexity and demanding significant manual labour (Bornhold, 1983; McKay et al., 2022). The advent of
Laser Particle Size Analysis (LPSA) has streamlined size-based IRD detection but introduces challenges such as small sample
volumes (<1 g) and limited control over contaminants (Hansen et al., 2015; Passchier, 2011; Passchier et al., 2021). Despite
their utility, the varied size definitions and methodological protocols lead to concerns about possible inconsistencies between
studies (Grobe, 1987; Kanfoush et al., 2000). This lack of standardisation has fuelled ongoing debates about the comparability
of results (McKay et al., 2022).
In practice the longest ranging IRD records can be feasibly reconstructed from marine sediment cores recovered through
scientific ocean drilling initiatives, such as the International Ocean Discovery Program (IODP; 2013–2024) and its
predecessors – the Integrated Ocean Drilling Program (IODP, 2003–2013), Ocean Drilling Program (ODP, 1985–2004), and



Deep Sea Drilling Project (DSDP, 1968–1983). The legacy cores of the IODP program family are curated in three repositories: the Gulf Coast Repository (GCR, USA), the Bremen Core Repository (BCR, Germany), and the Kochi Core Center (KCC, Japan). Collectively, these repositories house over five decades of marine sedimentary and stratigraphic archives, offering a unique opportunity to assess changes in ice-sheet behaviour and climate using legacy material. For instance, the earliest studies of Antarctic IRD from deep marine sites were conducted on DSDP Sites 268, 269, and 274, drilled during Leg 28 (December 1972 to February 1973). Piper and Brisco (1975) reported "large numbers of dispersed pebbles" in these Leg 28 cores, some embedded in deep-water pelagic sediments. Despite challenges in obtaining representative quantifications of IRD pebbles (e.g., Conolly and Ewing, 1965; Vorren et al., 1989), it was determined that analysing the sand-sized fraction is an effective method for identifying IRD. This approach, first applied to DSDP cores and subsequently to later ODP and IODP materials, builds on methodologies developed to study Arctic IRD around Alaska from cores recovered during DSDP Leg 18 (von Huene et al., 1973).

Innovative approaches emerged with the introduction of non-destructive scanning techniques for identifying dense IRD components, such as X-radiographs, 3D computed tomography (CT), and X-ray fluorescence (XRF). These non-destructive methods are increasingly favoured, particularly for IODP legacy cores with limited sample volume. Van Huene et al. (1973) pioneered using X-radiographs to identify IRD by visually counting pebbles in archive halves of cores and combining them with post-cruise X-radiography. Grobe (1987) refined this approach by incorporating whole-round x-radiographs for quantifying IRD >2 mm in 1 cm core intervals. Subsequent tests demonstrated the reliability of X-ray imaging methods compared to traditional manual counts, making it a preferred technique for assessing iceberg activity. Today, X-radiography is frequently complemented by 3D computed tomography (CT) scanning for enhanced precision. Recent advances also include automated IRD identification using convolutional neural networks (e.g., Jasper et al., 2024). However, these methods have only been recently developed and have yet to be adopted by the wider community.

In addition, XRF core scanning elemental data have also been shown to have potential use in the non-destructive identification of IRD-derived sedimentary layers, such as those formed during the Heinrich stadials of the last glacial (Hodell et al., 2008). These methods, however, are not applied often, as the small scan area of standard XRF core scanners limits the detection of dispersed or larger discrete IRD. To alleviate issues related to limited detector area, micro-XRF (µXRF) core scanning has recently gained popularity for providing high-resolution, non-destructive geochemical mapping. Detailed elemental analysis across the core section surface is achievable using instruments like the widely available Bruker M4 Tornado (20 x 16 cm capacity) and the newly developed M6 JETSTREAM (80 x 60 cm capacity). These 2D elemental maps offer insights into mineral composition and sedimentary variations (Feignon et al., 2021; Frery et al., 2021; Kaskes et al., 2021, 2024; Yang et al., 2018). These advances preserve core integrity while yielding critical data for provenance studies and IRD research (Bailey et al., 2012). The resulting geochemical maps allow for the differentiation between different sedimentary components, the identification of mineral phases, and the detection of compositional variations. Recent instrument advancement significantly





expands the scope of µXRF applications (see methods), as it is now possible to investigate entire core sections and other large-
scale geological samples. This large-area scanning capability is especially advantageous for ice-rafted debris (IRD) studies,
enabling efficient coverage of multiple core sections at high throughput. Several challenges remain in the accurate
identification of IRD, despite significant advancements in non-destructive IRD detection methodologies, including the lack of
standardisation across studies, the integration of automated approaches, and the limited capacity for simultaneous provenance
and geochemical analyses (Jasper et al., 2024; McKay et al., 2022). Here, we address these open questions by evaluating the
potential of integrating existing approaches with modern non-destructive techniques, such as µXRF imaging and k-means
clustering. We ultimately propose a multi-proxy non-destructive approach combining large-chamber µXRF and CT scanning
to provide a unified and efficient framework for IRD detection, differentiation, and geochemical characterisation in marine
sediment cores.

## 2 Materials and Methods

### 2.1 Site Location and Sampling during ReC23- 01

In 2023, the Repository Core Rediscovery Program (ReCoRD) program was initiated by a joint venture of the Kochi Core
Center (KCC), Kochi University, and the Japan Drilling Earth Science Consortium (J-DESC) as a new workshop type,
providing access to IODP cores archived at the KCC in Kochi, Japan. The first international ReCoRD workshop, ReC23-01,
'Tracing Intermediate Water Current Changes and Sea Ice Expansion in the Indian Ocean', was held between the 27th of
August and the 5th of September. 2023 at the KCC in Kochi (Auer et al., 2024).
One of the objectives of ReC23-01 was to gather new data to test the hypothesis that the expansion of sea ice around Antarctica
impacted water circulation in the Indian Ocean. To this end, several Indian Ocean legacy sites were re-described
sedimentologically and analysed with state-of-the-art techniques like CT and µXRF scanning. Site 266 was drilled during
DSDP Leg 28 between Jan 2nd and Jan 4th, 1973, in a present-day water depth of 4167 m (corrected echo sound depth). DSDP
Site 266 is located in the southernmost Indian Ocean between Australia and Antarctica (56°24.13'S, 110°06.70'E; Fig. 1;
Hayes et al., 1975), along the south flank of the Southeast Indian Ridge, about 800 km from the Ridge Crest. Site 266 was
initially targeted by Leg 28 to recover a Neogene sedimentary sequence unaffected by tectonic activity in the Indian Ocean
section of the Southern Ocean (Hayes et al., 1975). It is, therefore, an ideal paleoceanographic target for studying the northward
migration of the polar frontal system during the Neogene. Site 266 is also located at the northernmost extent of large Antarctic
iceberg concentrations (Romanov et al., 2017), so it is a key location to capture the inception of iceberg advance into the
present-day polar front. To test our non-destructive multi-proxy approach, we targeted three core sections of DSDP Site 266
(Sections 266-8R-3A, 266-9R-1A, and 266-9R-2A). These cores were selected based on limited shipboard biostratigraphic
information (Hayes et al., 1975) to fall in the interval after the Middle-Late Miocene re-expansion of Antarctica, but before



the larger Quaternary glaciations. This interval was chosen to ensure IRD was present, but not in significantly large amounts
that could be easily identified by eye, so that the efficiency of the non-destructive techniques could be properly tested.

**2.2 Micro X-ray Fluorescence Core Scanning and k-means clustering multivariate statistics**

For micro X-ray fluorescence core scanning (µXRF-CS), we scanned five archive half core sections from DSDP Site 266 with
a Bruker M6 JETSTREAM to at the Bruker Japan X-ray Division Demo Center (see Fig. 3). This device was specifically
selected as a test case, as the chamber dimensions of the more prevalent Bruker M4 Tornado and other similar devices restricts
the sample-size that can be analysed, limiting their application to marine drill core sections. The introduction of an open-beam
system pioneered by the Bruker M6 JETSTREAM means that larger samples can now be analysed as is, including standard
IODP cores. As the length of some sections was longer than the maximum measurement area of the M6 JETSTREAM (80 ×
60 cm$^2$), the upper and lower halves of the sections were measured separately for 1.5 m sections. The mapping resolution (pixel
size) was 500 µm × 500 µm, and scan speed was 5.5 ms/pixel for this machine demonstration. The X-ray source was set at 50
kV and 600 µA. A 1000 µm aperture management system was applied to account for surface irregularities. This system,
available in the analysis software, dynamically adjusts the aperture setting to provide a consistent beam diameter regardless of
the distance between the sample surface and X-ray source in case of surface irregularities. All measurements were done under
atmospheric conditions.
To objectively differentiate sediment types and anomalous grains in the µXRF datasets, we applied k-means clustering to the
elemental maps using the base kmeans() function in the R stats package (R Core Team, 2024). Prior to clustering, the µXRF
count data for each pixel were standardised (z-score normalisation) to ensure comparability across elements with different
count magnitudes. Clustering was performed using only those elements with sufficiently high and consistent signal counts
across all sections: Al, Ca, Fe, K, Mn, Ni, Pb, Rb, S, Si, Sr, Ti, Zn, and Zr. For each core section, we tested multiple values of
k. We visually evaluated the cluster outputs to ensure meaningful sedimentological interpretation, selecting k = 3 for 266-8R-
3A, and k = 4 for 266-9R-1A and 266-9R-2A, using 10 random starts and a maximum of 100 iterations. Clusters were then
visualised spatially to identify and isolate anomalous grains based on elevated Fe, Mn, and Zn counts. Full clustering code and
input parameters are provided in the supplementary material (Auer et al., 2025).

**2.3 CT-scanning and CT-based IRD quantification using the Waikato Environment for Knowledge Analysis (WEKA)**

To generate a CT-image-based IRD record for the three test sections, CT scans of the archive half-core sections were produced
at the KCC using a Hitachi Medical Corporation with a 310 µm/pixel spatial resolution. We note that the comparatively low
resolution of such medical-grade CT scanners limits IRD detection only to larger pebble grains of at least 600 - 1000 µm in
diameter.





Nine archive half-core sections were stacked together on the scanner bed for scanning operation and scanned simultaneously.
All sections remained in their D-tubes during this process. Individual core sections were subsequently automatically processed
into individual image sets by separating each core image along the respective D-tube of the core section. This methodology
guarantees optimal use of scanning time and provides fast processing of large quantities of the legacy core at the KCC. After
analysis, semi-automated brightness and contrast adjustments were applied to all CT-image stacks of the targeted sections to
ensure comparable density brightness relationships between stacks. The CT stacks were subsequently processed with Fiji
(ImageJ) (Schindelin et al., 2012), applying the trainable 3D-WEKA segmentation based on the Waikato Environment for
Knowledge Analysis (WEKA) environment (Arganda-Carreras et al., 2017; https://imagej.net/plugins/tws/). As implemented
in Fiji, the WEKA segmentation plugin provides a comprehensive set of open-source machine learning algorithms combined
with image tools in a graphical user interface. It has been implemented for both 2D and 3D image sets. The plugin offers the
option to define multiple classes depending on the needs of the study. Once the number of classes is determined, training sets
can be generated through the graphical user interface (Fig. 2a, 2b). Sets can be adjusted and refined depending on the outputs
of the machine learning classifier until a satisfactory output is reached (Figure 2c, 2d). Crucially trained classifiers can be
easily saved and loaded for later use. The output of the classifier can then be saved as a new image (or image stack) and used
for further analyses, such as thresholding, 3D rendering, or any required statistical analyses (i.e., calculation of pixel
abundances) using freely available statistical and image analysis tools in Fiji.
For training, we defined five classes based on key features and used them for the automatic identification of features in the
core images (Fig. 2). These selected features include voids (= air), the core liner, D-tube, sediment, cracks within the sediment,
and higher-density grains/components within the sediment. The results were quantified based on the sum of pixels assigned to
each class using the quantification tools in Fiji, based on $3.88*10^7$ classified pixels. Subsequent trial applications of the trained
WEKA classifier outside the selected target sections for µXRF scanning yielded mixed results (see discussion for details).
**2.4 Manual IRD Picking and Scanning Electron Microscopy-based Energy Dispersive X-ray Spectroscopy**
One test sample from Site 266 Core 9R-2W was washed and picked for IRD. The sample was freeze-dried, after which a split
of 7.274 g of dry sediment was wet-sieved through 125 µm and 63 µm mesh sieves. The recovered sediment fractions were
oven dried at 45°C for 24h. The 125 µm fractions were picked for IRD, and 19 potentially lithic grains and one fish tooth (Fig.
2), were picked and mounted on a scanning electron microscope (SEM) stub. All grains were imaged and measured using a
Keyence VHX-7000 digital microscope and subsequently coated with C-Pt (0.5nm and 3nm, respectively) using a Leica
ACE600 sputter coater for SEM analyses. SEM analyses were performed with a Zeiss Gemini DSM 982 at the University of
Graz equipped with an Oxford UltimMax 40 silicon drift energy dispersive X-ray spectroscope (EDS). The grains were imaged
using secondary electron detector imaging and subsequently analysed for their elemental composition using EDS analysis.
Quantitative elemental analysis, element mappings, and mineral phase classification were performed using the commercially
available Oxford AZtec® software suite.




## 3 Results

### 3.1 Micro X-ray Fluorescence Core Scanning



The clearest elemental signals in all analysed sections were associated with Ca and Fe, which is consistent with the lithological
composition of these sediments, dominated by a mixture of coccolithophore ooze and diatom ooze (Auer et al., 2024; Hayes
et al., 1975). Relatively low counts of Al and Si, which are nevertheless expected to be significant sediment components, can
be attributed to two factors. First, the measurements were conducted under atmospheric conditions, reducing detection
efficiency for lighter elements like Si and Al due to the greater absorption of low-energy X-rays by air. Second, elemental
counts in XRF data do not scale linearly with elemental concentrations, as differences influence them in fluorescence yield,
matrix effects, and atmospheric absorption. Notably, several localised zones enriched in Fe and Mn, occasionally accompanied
by Zn enrichment, provide evidence of anomalous grains within the otherwise pelagic sediment. A particularly prominent
feature is an anomalous grain identified in the upper portion of 266-9R-1A, measuring approximately 3 cm wide and 1 cm
long, with a distinctive co-enrichment of Fe, Mn, and Zn (Fig. 4).
A k-means multivariate approach enabled the 2-dimensional spatial classification of distinct sedimentary components (e.g.,
clay-rich, diatom ooze, carbonate ooze), as well as the identification of cracks and the highlighting of geochemical outliers,
such as anomalous grains potentially related to IRD. In 266-8R-3A, 266-10R-3A, and 266-11R-2A, the Fe distribution is
relatively homogeneous, with no significant anomalies detected by the clustering algorithm (Fig. 5a). In contrast, sections 266-
9R-1A and 266-9R-2A exhibit distinct localised zones of elevated Fe, highlighted by the contours. These zones correspond to
the anomalous grains identified during clustering analysis, particularly in 266-9R-1A, where the most prominent grain is
characterised by pronounced co-enrichment of Fe, Mn, and Zn. Thereby, Fe is the signature element for identifying anomalous
grains, which are interpreted as geochemical outliers within the pelagic sediment matrix in 266-9R-1A and 266-9R-2A. Figure
5 illustrates the distribution of Fe counts derived from micro-XRF scans for core sections 266-8R-3A, 266-9R-1A, and 266-
9R-2A, overlaid with contour plots delineating regions of anomalous grains based on the k-means classification.
In 266-8R-3A, the three delineated clusters corresponded to clay-rich sediment, diatom-rich sediment, and cracks (Fig. 6). In
this section as well as 266-10R-3A and 266-11R-2A (not figured), the analysis thus not identify any anomalous grains present
at the core surface. The analysis with $k$=4 in 266-8R-3A further subdivided the pelagic sediments without producing
meaningful new insights when focused on anomalous grain detection. In contrast, clustering with $k$=4 in sections 266-9R-1A
and 266-9R-2A yielded sedimentologically meaningful results (Fig. 6). Two clusters corresponded to different sediment types
(clay- and diatom-rich), one cluster captured cracks, and the fourth cluster identified anomalous grains with enhanced Fe, Mn,
and Zn signals. These results aligned with the observed grain described earlier in 266-9R-1A and confirmed its distinct
geochemical signature as a potential IRD-related feature.





**3.2 CT-scanning and WEKA Classification**

Applying the training set defined on Core 266-9R-1A to our four other sections resulted in five fully classified core sections (Tab. 1). The data is presented in percentage values following the individual assignment of all $3.88*10^7$ pixels. Raw data output for each analysed section can be found in the supplementary material (Auer et al., 2025). Crucially, the advantage of WEKA (or other machine learning) based algorithms is that the results include the brightness of each pixel (i.e., classical thresholding) and the overall image composition and pattern shapes of defined traits within training sets. It was thereby possible to distinguish cracks in the sediment from large voids (such as the space between the liner and the D-tube).

**3.3 IRD Picking and SEM-EDS analyses of grains**

The IRD identification based on 20 grains from sample 266-9R-2W 72/74 to groundtruth the validity of both µ-XRF and CT-scanning yielded surprising results. Of all 20 grains in the > 125 µm fraction, 17 are identifiable as angular to sub-angular lithic grains and can be classified as IRD. One grain was identified as the cusp of a fish tooth, and two were initially classified as "unknown flakes". Subsequent SEM-EDS analysis showed that identified IRD grains are angular to subangular grains of quartz, plagioclase, and albite. The grain size of picked IRD (n = 17) ranges between 244.93 and 805.23 µm (avg. 517.63; Tab. 2). One grain (IRD_02) consisted of a complex, pegmatitic Na-Feldspar (albite) with a complex accessory mineral assemblage including quartz, K-rich feldspar (orthoclase), rutile, apatite, and biotite. Another grain (IRD_15) consists of a complex Ni-Al with accessory amounts of Zn, Cu, Si, and Fe. Additionally, two of the picked grains display an angular flake-like nature, with a distinct reddish brown to black habitus. Closer analyses of these grains revealed they were composed of Fe, Mn, and V and their corresponding hydroxides and/or oxides (Tab. 3).

**4 Discussion**

**4.1 Trainable 3D WEKA-Image segmentation - A promising tool for IRD identification**

The output of the CT scanner resulted in some caveats underscored by the IRD-picking results. The image resolution of a Hitachi PRATICO medical CT scanner (310×310 µm per pixel) makes it impossible to detect smaller IRD (< 650 µm) based on the theoretical limitations due to detector aliasing. Conservatively, we thus assume that only grains of at least 1000 to 1500 µm can be detected consistently (Takase, 2024). This technological detection limit will naturally lead to a potential bias in the resulting IRD abundances and consistency in detection, especially since smaller size fractions (>125 to 1000 µm) of IRD may prove critical for defining IRD presence (and variability) in more distal sites in the southern hemisphere high latitudes.

Nevertheless, WEKA-based machine learning segmentation worked exceptionally well in the core sections used for training (266-8R-3A and 266-9R-2A). Complete and accurate classification could already be achieved by minimal training. The training involved manually classifying only 2-10 individual CT-image slices, which took less than 5 minutes each. All other



selected presented core sections were then successfully classified, yielding simple-to-extract and easy-to-process data (see
Tab. 1). In addition to consistently detecting high-density grains, our WEKA-based classification approach also allowed the
quantification of sediment alteration. For instance, we could quickly evaluate the quantity and position of cracks within the
studied legacy core material. However, issues were still encountered when WEKA training was applied indiscriminately
without basic sedimentological and drilling technological considerations. For instance, our presented training set failed to
classify the bulk sediment correctly when using it to core sections recovered at shallower drill depths (i.e., Cores 266-1R to
266-3R). While high-density grains were still identified consistently, the reduced sediment density upcore made it difficult for
the classifier to accurately identify sediment and associated structures (cracks/voids) throughout the test Site. Due to this
natural change in sediment dry bulk density downcore, it will likely be necessary to select discrete intervals based on dry bulk
density (DBD) thresholds to train several WEKA classifiers, should this approach be used in a larger context. Considering
WEKA's fast and easy training process, generating additional training sets tailored to specific DBD thresholds can easily
overcome this limitation. Evaluation of the effect of downcore DBD changes on WEKA-based image classifiers would have
been a critical next step if other issues (see below) had not precluded further advancement of this part of the study.

**4.2 Advantages and pitfalls of μXRF-based elemental analysis of IRD**
The analysed sections from DSDP Site 266 were selected to identify and characterise the geochemical signatures of ice-rafted
debris (IRD) through micro-XRF core scanning, leveraging its high-resolution, non-destructive capabilities. The anomalous
grains identified in sections 266-9R-1A and 266-9R-2W are notably enriched in metals, particularly Fe, Mn, and Zn,
distinguishing them from the surrounding pelagic (clay- or diatom-rich) sediment (Fig. 5). In this case, k-means clustering
successfully detected these grains when they were present. However, determining the optimal number of clusters (k=3 or k=4)
required some manual inspection to ensure the most meaningful results. This semi-automated approach demonstrates promise
in detecting anomalous grains, which supports the broader ambition of systematically identifying and chemically characterising
anomalous grains, such as IRD, in marine sediment cores. Here, the combined evaluation of segmented CT-scanning 3D image
information and the 2D elemental data of the M6 JETSTREAM has proven especially invaluable (Fig. 7).
However, some suspicion arises about the exact nature of the anomalous grains. Ideally, IRD should reflect a typical lithic
composition containing Si, Al, Fe, Ti, Mg, and similar common elements, with Mn only being a trace element. Therefore, the
consistent co-occurrence and strong signal in the μXRF results required an alternative explanation. One hypothesis could be
that these areas represent ferromanganese nodules or Fe/Mn crusts coating IRD or other lithogenic components, which could
be a distinct possibility in the study region (Albarède et al., 1997; Verlaan et al., 2004). Ferromanganese nodules and crusts
are known to contain other easily detectable elements such as Ti, Cu, Co, and Ni alongside Fe and Mn, albeit in lower
quantities. However, higher concentrations of these elements were not observed in the μXRF results.





As these accessory elements, however, usually occur only > 1 wt. % within Indian Ocean ferromanganese nodules (Albarède
et al., 1997), non-detection due to instrumental limitation could not be discounted a priori. It was, therefore, deemed necessary
to assess the chemical composition of potential IRD grains in the sediment using SEM-EDS. These analyses of all 17 IRD
grains from sample 266-9R-2W 74/5 (Fig. 8a) showed that the grains are of felsic composition (Tab. 2, Fig. 8b) and confirmed
that they do not have ferromanganese coatings. Conversely, the two additional "flake-shaped" grains that were encountered
and picked within the washed sediment showed high Fe and Mn concentrations indicative of Fe-Mn steel commonly used as
drill pipe material (Fig. 8a). These flakes were identified as Fe-Mn-V phases with a distinct metallic shine and showed signs
of possible Fe(III)-oxide-hydroxide alterations (Fig. 8c). The elemental composition of these two flakes are a close match with
the typical composition of API drill pipe steel (Godefroid et al., 2017; Ziomek-Moroz, 2012; Fig. 8c; Table 3), with small
compositional differences potentially explained by analytical imprecision and the possibility that the GLOMAR Challenger
used custom steel for its drill pipes. Using the EDS analysis, we can identify that the two recovered flakes were rusted pieces
of drill pipe steel that contaminated the sediment of Site 266. Drill pipe contamination had been noted by the Leg 28 scientific
party (Hayes et al., 1975). The ReC23-01 team also observed heavy contamination at core ends and the edges of the core liner
of both archive and working halves at Site 266. However, the authors were unaware of the extent of drill pipe contamination
within the body of the recovered sediment until this was revealed by later analysis.

### 4.3 Implications for non-destructive multi-proxy approach to identifying IRD in IODP/ODP/DSDP legacy cores

Our non-destructive multi-proxy approach combining CT and µXRF scanning results in a consistent framework for high-
resolution imaging of IODP legacy core material, with demonstrable value in rapidly identifying and chemically classifying
IRD (especially in the size range 1000-1500 µm). The µXRF scanner results were extremely valuable for the identification
and chemical characterisation of surface features in drill cores. The rapid analysis speed and excellent analytical resolution
(500×500 µm used in this study) are highly complementary to the applied CT scanning technique. Higher resolution analyses
would likely further enhance the data quality and compatibility with CT scanning data. A resolution of 100×100 µm is routinely
possible with the JETSTREAM, for instance. Furthermore, the quality of µXRF elemental analysis could be improved when
running the system under He-flush. This is expected to significantly enhance the signal quality for Si and Al, two of the most
critical elements when identifying anomalous grains such as IRD on cut core surfaces. Differentiation between drill pipe
contamination and IRD should be possible based on the density difference. Steel has > 7 g/cm³, while IRD is expected to show
a density range between 2.5 and 3.8 g/cm³. Once accurately calibrated, this could be done with available CT scanning data and
the applied WEKA machine learning tool. However, further work and µXRF data would be necessary to achieve this, as the
Site 266 core sections used for this study did not have large visible IRD grains present.. Hence, we could not unequivocally
differentiate between IRD and drill pipe contamination when selecting the key features for training the WEKA classifier.
Ultimately, our non-destructive multi-proxy approach represents a new benchmark for consistent high-resolution imaging of
sedimentary core material. Identifying drill-pipe contamination through µXRF and EDS analysis highlights that it remains





critical to ascertain the origin and chemical composition of high-density grains in legacy core material before attempting
meaningful sedimentological and paleoclimatological interpretation.
**5 Conclusions and Outlook**
For the first time, it has been possible to collect µXRF data on entire section halves using the open-beam system incorporated
into the Bruker M6 JETSTREAM in a way that was previously not possible due to limits in chamber dimensions. The µXRF
data can be used to identify surface elemental variability of sedimentary core surfaces, and is an immensely valuable dataset
for high-resolution characterisation of the chemical composition of sediment. For instance, here we could identify the Fe/Mn
signature of drill pipe and distinguish surface records of IRD from drill pipe contamination at DSDP Site 266.
Furthermore, we have shown that medical CT-scanning data of legacy core material holds unprecedented power for machine
learning-based sediment classification and segmentation. Although powerful, fast, and reliable, any such automated
classification still requires careful validation and ideally discrete testing of base assumptions. Here, we also wish to highlight
the power of µXRF core imaging as an independent fast verification tool that should be used with most CT-based classification
algorithms, especially for IRD detection over large (hundreds of meters or more) core material. This study provided 3 key
conclusions.
1.  µXRF and CT scanning data of legacy core material are highly complementary, fast, and reliable non-destructive
analysis tools that provide unprecedented information. IODP legacy core material stored in the three Core Repositories
especially stands to benefit immensely from its future application and adoption by the community.
2.  Already available machine learning classification and segmentation-based CT scanning, such as the trainable 3D
WEKA segmentation implemented in Fiji, is highly successful in consistently characterising sedimentary features,
such as density anomalous grains (e.g., IRD, drill pipe contamination) in legacy core material.
3.  Successfully applying machine learning algorithms requires high-quality control pre- and post-analysis. In situ µXRF
analysis of entire core section halves is now possible with technological advancements such as the Bruker M6
JETSTREAM. Such whole-section 2D µXRF analyses enable high-fidelity 2D elemental imaging capabilities and
provide the necessary data to ground-truth density-based CT-scanning analyses with fully correlatable surface
elemental information.
In conclusion, while ultimately unsuccessful in its original goals, our study provides an up-and-coming methodological
framework for future high-resolution CT scanning and elemental imaging-based analyses of legacy core material for IRD
quantification.



## Code and data availability

All data and supplementary text are available on Zenodo (DOI: 10.5281/zenodo.15543202)

## Supplement

All data and supplementary text are available on Zenodo (DOI: 10.5281/zenodo.15543202)

## Author Contributions

GA and DDV conceptualised the study. YK and AS carried out the CT scanning and µXRF data acquisition. GA carried out CT-image machine learning data analyses, IRD picking, and SEM-EDS analyses, and DDV and AS performed the µXRF data analyses. All authors were part of the ReC23-01 Science Party and contributed to data capture, analyses, and interpretation during and after the workshop. GA and DDV wrote the paper and drafted the figures with input from all co-authors.

## Financial Support

This research was funded by the Austrian Science Fund (FWF) [Grant DOI: 10.55776/P36046] awarded to GA. For the purpose of open access, the authors have applied a CC BY public copyright licence to any Author Accepted Manuscript version arising from this submission. Additional funding was provided by a NAWI Graz Visiting Award for High Potentials, awarded to GA. Funding for µXRF scanning was provided by the Japan Society for the Promotion of Science (JSPS) KAKENHI Grant (grant no. 22KJ149800) awarded to AS. AJD was funded by the Natural Environment Research Council (grant number: NE/W009366/1) and the Royal Society (grant number: DHF\R1\221091).

## Competing interests

The contact author has declared that none of the authors has any competing interests.

## Acknowledgements

This research used samples and data provided by the International Ocean Discovery Program (IODP) and its predecessors. We thank the DSDP Leg 28 shipboard science party, the crew, and the technical staff of D/V Glomar Challenger. We thank Manabu Mizuhira (Bruker Japan) for his assistance with µXRF measurement and analysis. This study was conducted under the Repository Core Re-Discovery Program (ReCoRD) ReC23-01 workshop, a collaborative program between Kochi Core Center and J-DESC.



**Team List: Author Affiliations and Emails (in alphabetical order)**

**Or M. Bialik**, University of Münster, Institute of Geology and Palaeontology, Münster, Germany (obialik@uni-münser.de)

**Beth Christensen**, Rowan University, Department of Environmental Science, School of Earth and Environment, Glassboro, NJ, USA (christensenb@rowan.edu)

**Tamara Hechemer**, University of Graz, Department of Earth Sciences, NAWI Graz Geocenter, Graz, Austria (tamara.hechemer@uni-graz.at)

**An-Sheng Lee**, National Taiwan University, Department of Geosciences and Research Center for Future Earth, Taipei, Taiwan; now at: Marine Core Research Institute, Kochi University, Kochi, Japan.

**Jing Lyu**, University of Münster, Institute of Geology and Palaeontology, Münster, Germany (j.lyu@uni-muenster.de)

**Theresa Nohl**, University of Vienna, Faculty of Earth Sciences, Geography and Astronomy, Department of Palaeontology, Vienna, Austria (theresa.nohl@univie.ac.at)

**Natsumi Okutsu,** Japan Agency for Marine-Earth Science and Technology, Institute for Marine-Earth Exploration and Engineering, Yokosuka, Japan (okutsun@jamstec.go.jp)

**Werner E. Piller**, University of Graz, Department of Earth Sciences, NAWI Graz Geocenter, Graz, Austria (werner.piller@uni-graz.at)

**Xabier Puentes-Jorge**, University of Graz, Department of Earth Sciences, NAWI Graz Geocenter, Graz, Austria (xabier.puentes-jorge@uni-graz.at)

**Jumpei Yoshioka**, National Institute of Polar Research, Research Organization of Information and Systems, 10-3, Midori-cho, Tachikawa, Tokyo, Japan (yoshioka.jumpei@nipr.ac.jp)



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



## Tables and Table Captions

**Table 1: Classification results of the WEKA segmentation based on 38849328 quantified pixels in each core section.** The last column presents the % abundance of IRD/Flakes classified pixels compared to the total sum of sediment-classified pixels, indicative of the percentage of IRD within the sediment.

| Section ID | IRD/flake | cracks | liner | D-tube | void | sediment | IRD/flake in sediment |
|---|---|---|---|---|---|---|---|
| 266-8R3 | 0.00% | 0.88% | 4.09% | 15.74% | 50.40% | 28.89% | 0.01% |
| 266-9R1 | 0.05% | 0.80% | 5.14% | 9.10% | 69.28% | 15.62% | 0.32% |
| 266-9R2 | 0.02% | 3.61% | 6.05% | 8.20% | 58.61% | 23.51% | 0.07% |
| 266-10R2 | 0.01% | 1.34% | 5.44% | 11.58% | 57.16% | 24.48% | 0.04% |
| 266-11R2 | 0.01% | 3.41% | 5.66% | 8.85% | 57.46% | 24.60% | 0.03% |

**Table 2: Length measurements and interpreted primary mineralogy (based on SEM EDS results) of all picked grains >125 µm in sample 266-9R-2W 74/75.** Please refer to fig. 8 for the image and measurements of all picked grains.

| ID | length [µm] | primary mineralogy (EDS) |
|---|---|---|
| *cusp* | 477.58 | carbonate fluorapatite |
| *IRD_01* | 636.32 | quartz+feldspar |
| *IRD_02* | 805.23 | pegmatite (see text) |
| *IRD_03* | 801.75 | quartz |
| *IRD_04* | 474 | quartz |
| *IRD_05* | 510.84 | quartz |
| *IRD_06* | 586.98 | quartz |
| *IRD_07* | 695.02 | Ca-feldspar |
| *IRD_08* | 445.57 | Na-feldspar |
| *IRD_09* | 553.5 | quartz |
| *IRD_10* | 429.87 | K-feldspar |
| *IRD_11* | 269.93 | quartz |
| *IRD_12* | 497.69 | quartz |
| *IRD_13* | 530.41 | quartz |
| *IRD_14* | 645.51 | quartz |
| *IRD_15* | 244.93 | Ni-Al alloy ($Ni_3Al$) |
| *IRD_16* | 281.67 | quartz |
| *IRD_17* | 390.52 | quartz |
| *flake_01* | 1793.98 | MnV-Steel |
| *flake_02* | 341.99 | MnV-Steel |




507 **Table 3: EDS quantification results of the drill pipe flake shown in Figure 6c compared to accepted values of American Petroleum Institute (API)**

508 **X46 steel** (Godefroid et al., 2017; Ziomek-Moroz, 2012)**.**

| Element | wt% | wt% Sigma | API X46 Steel (min-max wt. %) |
|---|---|---|---|
| O | 23.67 | 0.01 | / |
| Si | 1.68 | 0.01 | 0.00 - 0.30 |
| S | 0.24 | 0.01 | 0.00 - 0.15 |
| K | 0.11 | 0.01 | 0 |
| Ca | 0.5 | 0.01 | 0 |
| V | 0.11 | 0.01 | 0.00 - 0.15 |
| Cr | 0 | N/A | 0.00 - 0.15 |
| Mn | 1.31 | 0.02 | 1.30 - 1.40 |
| Fe | 71.95 | 0.05 | balance |
| Cu | 0.25 | 0.03 | 0.00 - 0.20 |
| Zn | 0.18 | 0.03 | 0.00 - 0.15 |
| Mo | 0 | N/A | 0.00 - 0.15 |
| Total | 100 | | |

509





**Figures and Figure Captions**

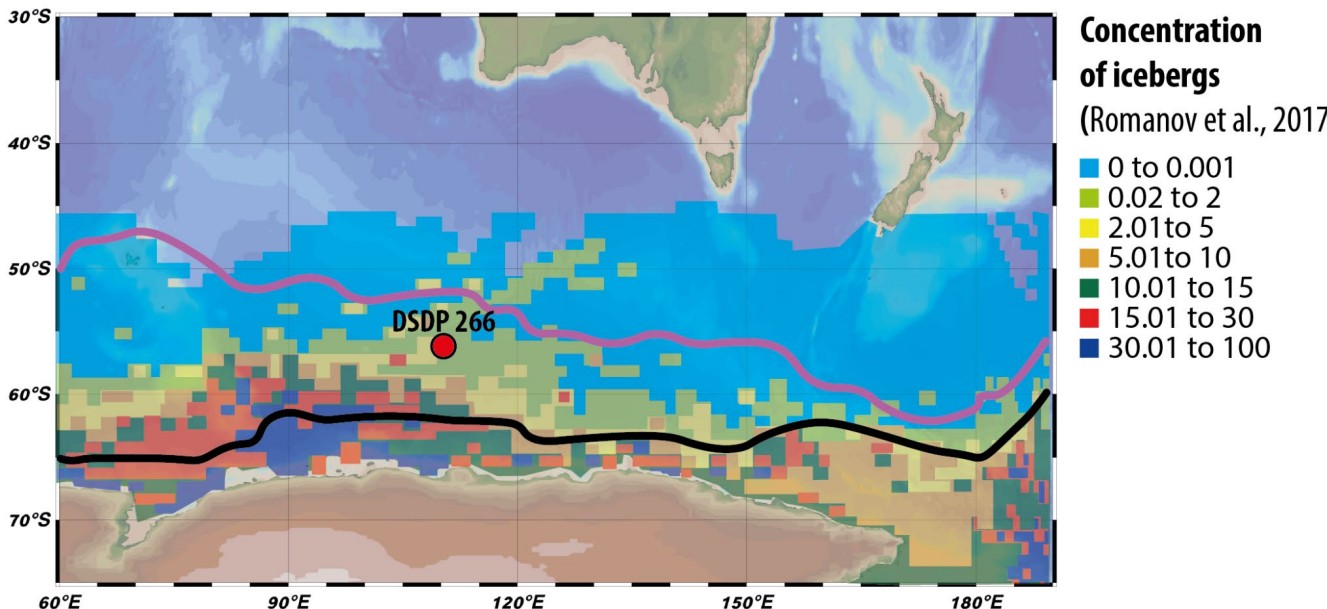

**Figure 1: Location map.** The figure shows the Position of Site 266 in the southernmost Indian Ocean (56°24.13'S; 110°06.70'E). The map further shows the average present-day position of the polar front (purple), the southern extent of the antarctic circumpolar current (black) and the present-day iceberg concentrations based on observational data from Romanov et al. (2017) in the area. The colour-coded concentration reflects the number of icebergs sighted within a 15-nautical mile range from the observing ship. Basemap was generated in OceanDataView (ODV v5.7.2; Schlitzer et al., 2021).







**Figure 2: 3D trainable WEKA segmentation and results:** Two WEKA training sets from section 266-8R-3A and 266-9R-2A (A and B, respectively). The six defined classes are (1) VOID (turquoise) = air/empty space; (2) D-TUBE (yellow) = the characteristic uniform density of the D-tube; (3) LINER (purple) = characteristic density of core liner; (4) CRACK (green) = visible cracks within the sediment; (5) SEDIMENT (pink) = sediment; (6) IRD (red) = high-density grains. WEKA classified all studied sections successfully, as shown through examples from Section 9R-1 (C) and 8R-3 (D). (E) flattened results of the WEKA algorithm of IRD/flake occurrences. The core image of section 266-9R-2A underlies the occurrences (vibrant orange). This way of processing WEKA results provides information similar to a standard X-radiograph.





**Figure 3: Foto from Bruker M6 JETSTREAM.** Y. Kubo loading the Bruker M6 JETSTREAM with a core section during the machine demonstration in August 2023 in Japan.



**Figure 4: Selected elements from MicroXRF Scan of DSDP Site 266-9R-1A.** Several anomalous grains of varying size emerge in the Fe, Mn and Zn data, marked by a white cycle in each element. Ca concentration and Fe concentration also reflect sediment variability. Mottling is due to heavy rotary drilling disturbance.





**Figure 5: Fe counts derived from µXRF scans.** Plots are overlain by a contour plot delineating anomalous grains. Red circles are added as a visual aid to show recorded contours in sections 266-9R-1A and 266-9R-2A.





**Figure 6: Classification of µXRF elemental data.** 266-8R-3A shows no anomalous grains, so the k-means classification algorithm only differentiates between clay-rich, diatom-rich sediments and cracks (k = 3). In 266-9R-1A and 266-9R-2A, the k-means classification algorithm produces meaningful results with k = 4, as the algorithm differentiates between clay-rich and diatom-rich sediments, cracks, and anomalous grains. High counts in Fe, Mn and Zn characterise the latter group.



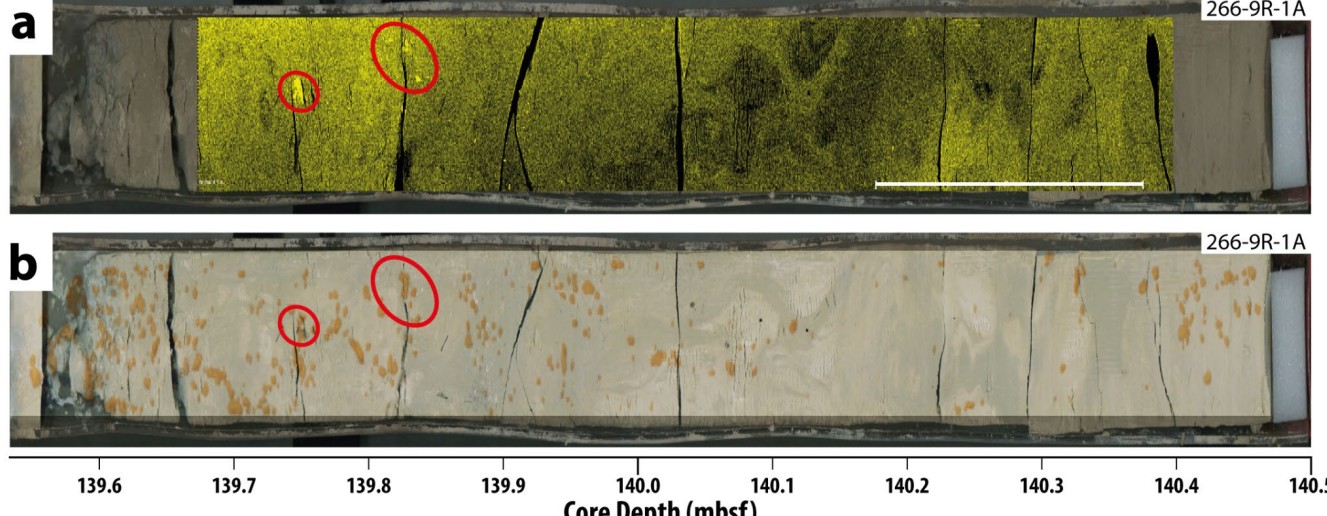

**Figure 7: Comparison of CT-image WEKA classification-derived IRD records.** Comparison of the µXRF derived Fe count intensity (yellow) with the 2D flattened occurrence cloud (orange) of high density grains based on the WEKA classifyer. **a**) visual core image and µXRF data and image information for section 266-9R-1A. Note the highlighted areas of high Fe counts indicating anomalous grains encountered at the surface of the core section; **b**) comparison of the line scanned core image and the 2D flattened WEKA IRD/flake segmentation output. the red circles indicate the ares of outcropping high density grains that are also clearly visible in the surface µXRF data shon in a).



**Figure 8: Picked IRD and visualized EDS results.** Note the different scales of each image. **a)** Composite photomicrograph of the 20 picked grains from discrete sample 266-9R-2W 74/75. White bars indicate the measurement axis for maximum length determination. White boxes indicate the area of EDS analysis shown in b) and c). Note the large >1.7 mm iron flake in the lower left of the picture. b) Example of the EDS-based elemental clustering-based phase identification using the Oxford AZtec® software suite on a complex pegmatitic grain. c) Aztec phase identification of the iron flake showing the covering of Fe-Mn oxide-hydroxides as well as less corroded metal surfaces that also contained appreciable amounts of vanadium (V), also not the stuck radiolarian shell that was classified as biogenic silica (opal).