# Peer review of "Automated Detection and Chemical Characterization of Anomalous"

_EGUsphere, 2025_

## Referee Comment (RC2)

[referee-annotated manuscript omitted]

---

## Author Comment (AC1)

The following reply represents a point-by-point response to the comments and observations of reviewer #1. We would like to sincerely thank the reviewer for their time and effort in commenting on and critically evaluating our manuscript. The supplied comments have been constructive in our efforts to improve the text of this manuscript. Our comments are indented and formatted in blue, while the original review comments are formatted in black and retain their original formatting.

Auer and colleagues present new methods for non-destructive study of clasts in marine drill cores, motivated by the need to study ice-rafted debris without excessive sampling of the limited amount of material. This work builds on many years of trying to determine the best approaches to do this and explores new methods focused on CT scanning and uXRF. It is a super interesting approach and I enjoyed reading about it—although I was not super convinced it would be the best approach for IRD quantification in these specific cores. That said, the science is sound, it is an interesting method, and promotes the use of legacy cores. I think it will be of interest to the wider community as they evaluate the best approaches for IRD quantification. I think it could be published with limited revision.

As a commentary—not a criticism of this work—in my own research, I have become more and more convinced that there is no better way to quantify IRD than breaking out the sieves and actually looking at what is there. For the record, this is coming from someone who initially really pushed for automated approaches. In my recent experience, we've found that other particles (like iron sulfides) can confuse many different detection algorithms and are responsible for the differences between x-ray/CT counts and physical sieved wt% data. In the author's work, metal flakes pose a similar problem. However, the datasets we've been developing from sieved samples have ended up quite different from x-ray derived methods in some intervals (supervised and unsupervised), with the sieved data seeming to make a lot more sense when compared to other proxy data. I was left wondering after Section 4.3 if the authors actually thought that these approaches were useful or if they would still require significant work looking at sieved fractions in order to ground truth their interpretations.

We thank the author for this assessment. Indeed, we agree that ultimately the most effective way to define IRD will always be through the washing of material and manual picking of IRD grains. However, this resolution, which is usually recovered by IODP expeditions, is neither feasible nor would it allow for the preservation of precious and often irreplaceable material, such as IODP legacy cores. It is thus imperative to explore alternative methodologies and evaluate their feasibility – and most crucially, report on the findings in a transparent and FAIR manner, irrespective of success.

We also agree that the statements found in Section 4.3 may benefit from clarification. We will therefore revise this section, also in accordance with the comments of reviewer #2, to clarify these points further.

As a final note, I wonder if the authors might comment a bit more about the metal flakes and if this is an issue that is likely limited to early DSDP rotary cored un-lithified sediments and to what extent the issue might persist in the more common legacy APC cored sediments. From my experience, I had thought that for APC cores, these flakes are generally limited to the near the core liner and generally in greater quantities near the top of the cores—and thus a bit easier to work around. I know that there has been a bit of documentation about this from the paleomagnetic community, with one of the primary reasons for the development of the 'u'-channel method to sample the more pristine centers of the cores to get away from

the metal flakes that were sometimes found near the core liner and messed up the half-core pass-through measurements (see the first u-channel study: Mead et al., 1986; https://doi.org/10.1029/PA001i003p00273). There is also documentation on rust contamination from Carl Richter's 2007 "Handbook for Shipboard Paleomagnetists." As the author's conclusions do not mention this complication, despite being a major caveat, I wonder if they think that their approach here might be better suited for legacy APC cores than legacy rotary cores (for soft mud), like reported here?

> We sincerely thank the reviewer for this insight. We fully agree that APC coring will likely be much less affected by such contamination issues, although exceptional cases may always exist. One such example is also ODP Site 707, which has suffered from severe drill pipe contamination during APC drilling, resulting in a strikingly cyclic magnetic susceptibility signal with an exact one-per-core (or ~ 9.5 m) periodicity.

> We also agree that these aspects should be discussed in more depth and therefore intend to add a relevant paragraph to our discussion at the end of section 4.2, stating:

> *"Although severe pipe flake contamination as found in this study is likely primarily associated with older DSDP material, similar iron contamination was also documented in newer ODP cores. For example, ODP Site 707 (located in the western equatorial Indian Ocean) exhibited pervasive contamination from pipe material. However, the use of the advanced piston corer (APC) limited severe contamination to sections 1 and 2 of each core. The Shipboard Scientific Party (1988) of Leg 115 notes a fine and relatively uniform distribution of the material severely affecting magnetic susceptibility and paleomagnetic analyses at Site 707. It is consequently not implausible that other ODP and possibly even newer IODP cores may pose similar challenges for non-destructive methods, such as µXRF and CT scanning, as shown here. As this has posed a common problem for paleomagnetists over the history of IODP methods, techniques like u-channeling have been developed to alleviate such contamination (Mead et al., 1986). We speculate that, at least for the higher-resolution µCT analyses, u-channeling may also serve as a viable alternative for newer APC-drilled OPD and IODP materials.*

> *In contrast, older DSDP material would not benefit from such an approach. The low sample surface area, however, would invalidate any benefit from large-area µXRF scanning methods, as were applied here. In this context, our findings highlight the importance of thoroughly assessing sediment quality before analysis. Any such assessments should include reviewing shipboard reports, relevant publications, visual core descriptions, and conducting spot sampling to ensure reliable interpretations from non-destructive imaging and elemental analysis methods."*

Minor comments:

Line 84: You do such a good job reviewing the literature, I am surprised you don't list some examples of CT scans being used to quantify IRD at the end of this sentence, including many studies that have previously developed automated detection routines from CT data.

> Thank you for these suggestions. Indeed, our review was primarily focused on the application of medical CT imaging on legacy IODP core material. We acknowledge

that this narrow focus may have resulted in an unintentionally skewed view on the application of CT-scanning data (especially in the field of micro CT imaging) that has been published based on other (generally shorter and fresher) core material. We will include a relevant extension to the introduction, in accordance with the suggestions of reviewer #2 (please refer to the response to reviewer #2 for details on the changes we intend to implement).

Line 181-182: Figure call out to Fig. 2, but it doesn't seem to align with the statement which seems to be suggesting that you are going to show a picture of the 19 grains and 1 fish tooth.  Please clarify this statement.  Or is it meant to call out to Figure 8?

Thank you. We will remedy this error and change the text to reference the correct table (not figure) 2, which we intended to call out here.

Figure 4 caption: Should cycle by circle?

Indeed, ‚circle‘ is correct. We apologize for the confusion and will correct the caption accordingly

Figure 7: spelling? Shon -> shown

Thank you for catching this typo. We intend to fix it in our revised version

Line 304-307: This would be true IF the object filled an entire voxel.  Based on the resolution of the CT scanner, it is unlikely that the flake would be large enough and the voxels would be an integration of the high density steel and matrix sediment (thus, more likely in the range of a lithic clast).  Unless the flakes were sufficiently thick?

Thank you for bringing this up. We fully agree and apologize for not being clear enough in our initial version of the manuscript regarding this critical methodological detail. We intend to clarify this in more detail in the revised version, as it has led to some confusion in revision #2 (please also see our response to their comment, which includes our intended revision, thank you).

---

## Author Comment (AC2)

The following reply represents a point-by-point response to the comments and observations of reviewer #2. We would like to sincerely thank the reviewer for their time and effort in commenting on and critically evaluating our manuscript. The supplied comments have helped us immensely in our efforts to improve the text of this manuscript. Our comments are indented and formatted in blue, while the original review comments are formatted in black and retain their original formatting.

To whom this may concern,

The manuscript by Auer et al. complements a growing number of studies that seek to advance the potential of high-resolution scanning methods to automate counting and classification of particles with paleoenvironmental relevance. But while there are interesting aspects to this work, notably the use of XRF systems that scan the entire surface of a core (rather than just a line), but also the application of plug-ins that allow machine learning classification of particles, I believe that there several major issues in the design of this study require reframing and major revision:

> We thank the reviewer for their considerable insights, which will undoubtedly help strengthen the clarity of a revised manuscript. However, we wish to generally emphasize that most of the structural issues identified by reviewer #2 are rooted in a profound misunderstanding of the technical and material limitations of IODP legacy material – hence why this study was designed the way it was and ultimately is presented as a test case for a multi-proxy framework to avoid false positive detections in CT-data by using µXRF elemental fingerprinting.

*Ground-truthing*: Throughout the manuscript, the authors highlight the need for ground truthing of (semi)automatic approaches like those presented here. Yet it appears that few aspects of their study have been ground truthed. For example, just one sample was IRD-counted using traditional (wet-sieving) approaches, out of three core sections. Also, no information is provided on the provided on the bedrock provenance of IRD in the analyzed sediments, making it hard to contextualize the XRF and SEM data. In this respect, I also noted that the former uses Fe and Mn as diagnostic indicators, while the former reveals a more diverse composition where both these elements do not always dominate.

> We appreciate the deep concern the reviewer has about these issues, with which we can wholeheartedly empathize. However, several aspects are left out of the reviewers' consideration when making their argumentation:
>
> 1) The distance of the recovered material to the source area of IRD on Antarctica is too large to make any a priori assumptions based on provenance (the reviewer should be able to reach these conclusions themselves, considering the distance of the site to Antarctica shown in Fig. 1). IRD at this position therefore feasibly includes all rock types encountered on the Antarctic continent.
>
> 2) One sample was sufficient to confirm the suspicion that Fe/Mn-defined "anomalous grains" found by the archive half surface µXRF scan are, in fact, drill pipe contamination, inherently hampering any further efforts at non-destructive IRD identification and chemical classification based on our available test cores.
>
> 3) Our application of µXRF and EDS analyses is completely misrepresented in this comment. Mn and Fe were not used as IRD indicators a priori. K-means clustering was

found to be a diagnostic tool to indicate instances of anomalous grains on the cut archive half surface of two core sections. Our EDS analyses were subsequently employed to identify the nature of these Fe-Mn phases on discrete instances of picked IRD. This SEM-based EDS analysis was necessary to confirm that the Fe-Mn signals originated from drillpipe Mn-V steel contamination in the sediment. In our opinion, there is no ambiguity or uncertainty in this matter.

Therefore, similar concerns of our team as raised by rev #2 led to the decision to initially use only a single sample for discrete IRD investigation. This single sample was enough to confirm the suspicion that all surface detections of anomalous grains by the µXRF scanner are indeed pipe flake contamination. Furthermore, the uniformly high CT density (within the caveats of detector aliasing) of all larger discrete objects within our test cores indicates that most (if not all) larger anomalous components in the sediment are likely pipe flake contamination. Consequently, the µXRF results were indeed picking up discrete and anomalous components of a specific chemical composition.

Therefore, our study is a clear example of how easily false assumptions can be made a priori, which may lead to erroneous results without careful testing and confirmation of the null hypothesis. In our case, the null hypothesis (that discrete particles of different density are IRD) was first falsified by µXRF scanning (revealing a chemical signature outside a typical lithic IRD) and later confirmed by discrete analyses of picked grains from the sediment.

We made every effort to clarify these points further in our revised manuscript (for details on how this was implemented, please refer to our responses to specific comments below).

*Classification*: while I am not an expert in the applied classification schemes, the fact that the metal flakes deriving from drill pipes were not clearly distinguished based on their distinct geochemistry (measured with XRF) and density (measured with CT), does raise questions about the way they have been applied here.

Here, we feel the need to clarify, as the reviewer seems to have misunderstood our conclusions: Metal flakes were clearly distinguished and identified. However, no instance of IRD was identified. The only detected grains on the core surface were clearly identified as Mn steel flakes. In essence, no lithic grains large enough to be detected by the used medical CT scanner (Hitachi PRATICO) or the µXRF scanner (M6 JETSTREAM) were present in the core material.

Finally, we wish to point out that it is very dangerous to assume that (medical) CT-scanning-based density information should be similar to µCT data. Due to the inherent limitations of the X-ray detection array, smaller grains may appear to have lower apparent densities because of signal aliasing within the detector grid, as dictated by the Shannon-Nyquist sampling theorem. This effect particularly affects particles within the range of 1 mm or smaller (basically all picked IRD grains and most smaller pipe flakes). Hence, differentiation of densities is not as simple as the reviewer proposes here and requires further work. We also intend to clarify this within our text, as this was apparently not made clear enough in the present iteration of this manuscript.

*Resolution*: While the authors certainly do not hide this fact, the major offset in resolution between their XRF and CT data makes it hard (impossible) to inter-compare both datasets.

Indeed, while they argue that their material did not contain IRD that could be easily identified by eye to test the potential of their scanning techniques, the used scanner can only resolve particles larger than 1000 micrometers (medium sand: visible). This is twice as high as the resolution of their XRF scans, although it is casually mentioned that these could have been generated at a 100-micrometer resolution.

> Again, we strongly disagree with the reviewer's assessment. Comparison of the data on a pixel-by-pixel basis is not a prerequisite for correct data interpretation, and it is undoubtedly, as also evidently shown within our manuscript, not impossible.
>
> The argument for using eye identification for IRD is also valid. However, this limits inspection to only the surface sediment of the archive half (as these are generally not allowed to be sampled or disturbed due to their status as permanent IODP legacy archives). Consequently, both visual and µXRF are limited to the same planar view, with µXRF having the clear advantage of providing elemental information of material often covered with a sedimentary matrix (hence why the visually identified IRD ultimately turned out to be an iron pipe flake covered by some additional sediment).
>
> We thus wish to emphasize to the reviewer that their line of argument completely ignores the material limitations and special conservational considerations necessary when working with IODP legacy core archive halves. These cores are exceedingly precious, and any destructive investigation should ultimately be avoided if possible. Analyses of working half samples are certainly possible – and were also performed during this study. However, only one sample had already revealed pipe flake contamination, leading to a new set of conclusions that we then extensively discuss in sections 4.2 and 4.3 of our manuscript – something the reviewer possibly overlooked when making this argument.

*Novelty*: the above issues with ground-truthing and resolution restrict the added value of this work, compared to existing studies that have, for example, automatically counted IRD particles in the fine sand fraction, while also ground-truthing these profiles against traditionally measured datasets, and harnessing the 3-D potential of CT data. What does not help either is that quite a few of these studies are not mentioned or cited by the authors (also see my in-text comments). In addition, the authors sometimes revert to the use of rather hyperbolic statements that feel a bit out of place, considering the above.

> Here, we again need to emphasize that the reviewer may have failed to consider the specific consideration of IODP legacy material or the technical aspects of the gathered data. Why the reviewer believes the 3D potential of CT data was not leveraged is somewhat baffling, considering that data classification and subsequent segmentation were performed on orthoslices. In other words, 3D visualization is not a requirement for 3D analyses.
>
> We appreciate the reviewer's suggestion of additional studies for us to cite and will ensure that we include any relevant work mentioned throughout their review annotations.
>
> Finally, we would like to reiterate that our initial intention was to focus primarily on the application of these methodologies to IODP material, as it requires special considerations due to its age and curation style. This may not have given the breadth

of work in this field adequate homage. We will expand the introduction accordingly in our revised version in the following way:

*"However, data analyses on μCT resolution for IODP samples have so far been limited by the requirements of vertical alignment and constant rotation of the sample material during image acquisition (Cnudde and Boone, 2013; De Vleeschouwer et al., 2023; van der Bilt et al., 2021), which preclude their application for whole section archive half analyses of IODP legacy material section halves due to their large size (1.5 m length) and often loose and fragile core material, which precludes tilting of the already cut core sections. For the analysis of IODP material, medical CT scanners thus remain the preferred 3D-imaging tool, despite lower resolution and reduced X-ray brilliance (Cnudde and Boone, 2013; Hodell et al., 2017)."*

In addition to these main/major concerns, I have provided detailed in-text comments in the manuscript, and hope that these will prove useful to improve this manuscript.

We sincerely thank the reviewer for their extensive comments on the manuscript text. What follows is a point-by-point response to each of these comments referencing the manuscript line(s) they were made in:

Lines 16 - 17 comment states: "General comment - some sentences, like this one, could do with a bit of tightening"

We thank the reviewer for this insight, and we will revise our text accordingly.

Line 21 comment states: "I will return to this later, but while CT has the potential to analyze objects like IRD grains in 3D, it appears that this study relies on 2-D slices. Hence, this is a bit confusing/misleading."

We apologize for the confusion. The WEKA-based image segmentation analysis was performed, as is the norm, on all 2D slices. However, the medical CT scanning data used is inherently 3D data, given it's a sequential image stack of up to 3101 image slices.

Line 28 comment states: "I will also get back to this later, but based on what I read, I find there is very little ground truthing here: just one IRD was verified using non-scanning (SEM-EDS) data, and there is no comparison against manual count profiles, for example."

Here, we need to emphasize the nature of the material we analyzed. IODP legacy material archive half sections are generally not allowed to be sampled. Hence, a direct comparison of CT output and manual counts is impossible. The available sample was taken from the corresponding working half as the next best approximation. As the first sample analysed from a working half already confirmed that pipe flake contamination was present, further destructive analyses of samples were deemed unnecessary, as the CT results would likely be affected by them. Density differentiation of smaller flakes will be further complicated due to signal aliasing when considering the detector limitations of a medical CT scanner.

Lines 64 – 66 comment states: "General comment: it would not hurt to tighten the text a bit. Lists like these are a bit exhaustive, and (in my opinion) not needed for the story."

We respectfully disagree with the reviewer's assessment. This is a key piece of information that also helps understand the evolution of the program and reflects the long history of material available through it, and thus the narrative of the manuscript. We therefore prefer to retain this section of the text.

Line 73 comment states: "Maybe nice for lay people to relate that to the 63 micrometer cut-off you refer to in the previous pararaph?"

Excellent point. Thank you for this suggestion. We will do so.

Line 79 comment states: "Think that is just one aspect. the tantalizing prospect of a (much) higher resolution, as well as a major reduction in (human) resource use, are others that ought to be considered."

Excellent point. We alluded to this throughout the text, and it will undoubtedly be beneficial here, so we intend to include these suggestions. Thank you!

Lime 84 comment states: "I think it would be considerate to actually cite this work here. Certainly add:

Cederstrøm, J. M., van der Bilt, W. G., Støren, E. W., & Rutledal, S. (2021). Semi-automatic Ice Rafted Debris quantification with Computed Tomography. *Paleoceanography and Paleoclimatology*, e2021PA004293. https://doi.org/https://doi.org/10.1029/2021PA004293

Ekblom Johansson, F., Wangner, D. J., Andresen, C. S., Bakke, J., Støren, E. N., Schmidt, S., & Vieli, A. (2020). Glacier and ocean variability in Ata Sund, west Greenland, since 1400 CE. *The Holocene*, *30*(12), 1681-1693.

Hodell, D. A., Nicholl, J. A., Bontognali, T. R., Danino, S., Dorador, J., Dowdeswell, J. A., Einsle, J., Kuhlmann, H., Martrat, B., & Mleneck-Vautravers, M. J. (2017). Anatomy of Heinrich Layer 1 and its role in the last deglaciation. *Paleoceanography*, *32*(3), 284-303."

We thank the reviewer for these suggestions and will implement them accordingly. Additionally, we plan to expand the text to provide a more detailed account of the history of CT-scanning data in the field.

Lines 85 – 86 comment state: "I don't see why this is a "however"."

We will revise this sentence accordingly. Thank you for the suggestion

Line 91 – 99 comment states: "Far too much detail for an introduction: tighten please, and just keep the details in the methods section."

We politely disagree. This technology is new enough that a short discussion in the introduction is warranted, in our opinion. We nevertheless intend to reduce the length in our revised version, as it has indeed become quite long and includes some aspects that are better suited for the methodology section.

Lines 100 – 103 comment states: "I think that many (the majority) of these statements do not pertain to the cited studies - who have made significant strides forward to allay them. Given your results, i would tone this down, to avoid the impression that this work raises the bar

further than these studies. Also, again, i think important work is just not cited here, which - given the modest size of this field - is very surprising. In the context of automated counting using CT in 3D, the work by Cederstrøm et al. should certainly be acknowledged:

Cederstrøm, J. M., van der Bilt, W. G., Støren, E. W., & Rutledal, S. (2021). Semi-automatic Ice Rafted Debris quantification with Computed Tomography. *Paleoceanography and Paleoclimatology*, e2021PA004293. https://doi.org/https://doi.org/10.1029/2021PA004293"

In Line 103, an additional comment also states: "Again, far too strongly stated - you are not starting with a clean slate, as you appear to imply. Please, tone this down."

> We thank the reviewer for these insights; however, the core statement remains, and the in tandem approach with 2D elemental analyses of the split archive half surface has not been demonstrated in any studies known to us, including those mentioned by the reviewer. We will revise our introduction in the following way:

> "*Today, X-radiography-based analyses are frequently complemented or replaced by 3D computed tomography (CT) scanning for enhanced precision in density-based identification of anomalous grains and layers (Cnudde and Boone, 2013; Hodell et al., 2017; Røthe et al., 2018; van der Bilt et al., 2018). Here, in particular, the use of micro-computed tomography (µCT) imaging has been successfully applied for quantifying Late Pleistocene to Quaternary ice-rafted debris in core material from the northern high latitudes in marine and lacustrine sediments (Røthe et al., 2018; van der Bilt et al., 2018; Cederstrøm et al., 2021). Advances in µCT scanning have even progressed to the point of estimating grain size distributions within sediment samples (Auer et al., 2025). However, data analyses on µCT resolution for IODP samples have so far been limited by the requirements of vertical alignment and constant rotation of the sample material during image acquisition (Cnudde and Boone, 2013; De Vleeschouwer et al., 2023; van der Bilt et al., 2021), which preclude their application for whole section archive half analyses of IODP legacy material due to sample material limitations. For the analysis of IODP material, medical CT scanners thus remain the preferred 3D-imaging tool, despite lower resolution and reduced X-ray brilliance (Cnudde and Boone, 2013; Hodell et al., 2017).*"

> And also:

> "*Here, we address this methodological gap by evaluating the potential of integrating existing approaches with modern non-destructive techniques for chemical analyses, based on µXRF imaging and k-means clustering. We ultimately propose a multi-proxy non-destructive approach combining large-chamber µXRF and CT scanning to provide a unified and efficient framework for IRD detection, differentiation, and geochemical characterisation in marine sediment cores.*"

Lines 112 – 114 comment states: "Another example where, in my opinion, too much detail is provided. Tighten."

> Providing information on the program and its background context is critical, in our opinion. We would thus prefer to retain this information to show the study's place within its larger context.

Line 124 comment states: "Might wanna reference your figure, which shows this."

> Agreed. We will do so. Thank you for pointing this out.

Line 125 comment states: "Length??"

> Excellent point, thank you. We will add the length information to the individual core section identifiers listed in Line 126 within the original draft.

Line 129 1st comment states: "Adding this comment upon re-reading, after going through the entire paper: this sits awkwardly with the fact that the resolution of your CT data is not suited to identify particles that are not visible to the naked eye."

> We acknowledge that this may not have been phrased clearly enough. We meant "by visual identification on the split surface of the archive half section". We will revise this accordingly:

> "*This interval was chosen to ensure IRD was present, but not common enough that it could be visually detected on the split surface of the archive halves, (...).*"

Line 129 2nd comment states: "This sits awkwardly with the stated focus on "ground truthing"."

> We intend to revise this as follows: "*(...) to test the viability of the selected non-destructive methods.*"

Line 138 comment states: "Relevant to specify what source was used, to better get a idea of how well certain elements were measured."

> We will specify that the Bruker M6 JETSTREAM used in our study is equipped with a Rh anode microfocus X-ray tube (operated at 50 kV and 600 µA). We will also clarify that the measurements were conducted under atmospheric conditions, which results in the highest data quality for mid- to heavy elements such as Ca, Fe, Mn, and Zn.

Line 143 comment states: "I think a short explanation of, and justification for, k means clustering would not harm here."

> In the revised manuscript, we intend to add a brief justification of our use of k-means clustering. We clarify that this unsupervised multivariate method was chosen because it highlights 2D areas of similar elemental composition in the µXRF maps, allowing objective and reproducible grouping of pixels into sedimentary components and geochemical anomalous grains.

> "*We selected k-means because this unsupervised multivariate approach highlights contiguous 2D areas of similar elemental composition in the core scans, thereby reducing subjectivity compared to manual thresholding and enabling reproducible separation of background sediment types from geochemical anomalous grains. Clustering was restricted to elements that consistently produced reliable signal intensities above background noise across all analysed sections. This excluded light elements (e.g., Mg, Na) whose low fluorescence yields and absorption in air resulted in undetectable signals. The final element set (Al, Ca, Fe, K, Mn, Ni, Pb, S, Si, Sr, Ti, Zn, and Zr) include both lithogenic (e.g., Al, Si, Ti, Fe, K, Rb) and*"

*biogenic/carbonate-associated components (e.g., Ca, Sr, S), as well as trace metals (e.g., Mn, Zn, Ni, Pb, Rb, Zr).*"

Line 146 comment states: "Best report how this was determined / assessed. Now vague."

In our revision, we intend to clarify that clustering was restricted to elements that produced reproducible and stable signals across all core sections. Lighter elements (e.g., Mg, Na) were excluded because of their low fluorescence yields and strong absorption in air, which resulted in highly variable or undetectable counts under our measurement conditions. The chosen set of 14 elements (Al, Ca, Fe, K, Mn, Ni, Pb, Rb, S, Si, Sr, Ti, Zn, Zr) consistently showed sufficient signal-to-noise ratios and covers lithogenic, carbonate/biogenic, and trace-metal components, making them suitable for robust clustering.

Line 148 1st comment states "What...does this mean, exactly? You state earlier on, that you choose the sections so that they would contain IRD; but not in a way that could be visually counted. So, what was evaluated, and against what, in this case?"

Please refer to our response to your 2nd comment at Line 148 below, as it covers this issue as well.

Line 148 2nd comment states: "Also unclear how you ended up with this number of clusters for each section. I reckon this was (also) based on the dataset variance explained by the clusters? can you please elaborate on this?"

In the revised manuscript, we intend to clarify how the choice of k was assessed. In Section 266-8R-3A, a clustering solution with k = 4 yielded a geologically meaningful separation between cracks, clay-rich sediment, diatom-ooze-dominated sediment, and anomalous grains. In contrast, for Sections 266-9R-1A and 266-9R-2A, the higher carbonate content reduced the geochemical contrast among pelagic sediment types. Here, only three robust classes could be distinguished: background marine sediment, anomalous grains, and cracks. We intend to update this section of our manuscript to explicitly describe these criteria for determining the number of clusters in the following way:

"*In Section 266-8R-3A, clustering with k = 4 clearly differentiated cracks, clay-rich sediment, diatom-ooze dominated sediment, and anomalous grains. In contrast, in Sections 266-9R-1A and 266-9R-2A, higher carbonate contents blurred the geochemical separation between the diatom-dominated and clay-dominated sediment types. Hence, meaningful clustering was only achieved by distinguishing three classes (k = 3): pelagic sediment, anomalous grains, and cracks. Accordingly, we selected k = 4 for 266-8R-3A, and k = 3 for 266-9R-1A and 266-9R-2A, using 10 random starts and a maximum of 100 iterations.*"

Line 150 comment states: "So, information from the other elements was not diagnostic? Best specify why only these three were considered."

In the revised manuscript, we intend to clarify that clustering was initially performed using a set of 14 elements that consistently produced reliable signal intensities across all sections (Al, Ca, Fe, K, Mn, Ni, Pb, Rb, S, Si, Sr, Ti, Zn, Zr). However, when visualising the cluster results spatially, only Fe, Mn, and Zn proved diagnostic for

isolating anomalous grains in our test cores. These elements consistently co-occurred as enrichment "hot spots" against the pelagic background sediment. In contrast, other elements (e.g., Al, Si, Ca) primarily reflected the bulk matrix composition and thus did not contribute to the discrimination of discrete grains. We have revised Section 2.2 to make explicit that Fe, Mn, and Zn were not chosen a priori, but rather emerged as the most informative tracers during the clustering analysis.

Line 154 comment states: "Just the brand, no model is specified. Likewise for settings: as with xrf scans, one ought to report the voltage and current used. in addition, maybe filters were used too? All missing now."

> We apologize for the formatting error. The correct type of scanner is stated in Line 240, though. We intend to add the following: "(…) *using a Hitachi Medical Corporation PRATICO medical CT-scanner with a 310 µm/pixel spatial resolution with 16-bit grayscale values. Each core section was imaged sequentially as individual 3101 frames, one taken every 500 µm, resulting in a voxel size of ~0.048 mm$^3$.*"

Lines 154 – 156 comment states: "On a positive note - you are honest about this limitation. On a less positive note, it severely restricts the added value of this work. First and foremost, because other studies (see above) have used far higher resolution CT scans to count the sand-sized IRD particles you also allude to in your introduction. The difference in resolution with these studies is not marginal, but an order of magnitude. In addition, the CT scanning resolution is also much lower (and i will get back to this later) than your XRF scan results, limiting the complementarity of both approaches."

> While we agree with the reviewer on the technical limitations (as stated in our initial manuscript), material limitations necessitate inevitable trade-offs when dealing with large quantities of somewhat delicate material. To our knowledge, there is no flat-bed µCT scanner available, and tilting the archive halves is too dangerous, as sediment may shake loose, disrupting the recovered sedimentary succession. The second comment about the different resolution between CT and µXRF scans, we disagree with. No reason running analyses on different resolutions of the same material should prevent meaningful interpretation. Of course, a one-to-one comparison of each pixel/voxel will not be meaningful, but that was never the intention of our geochemical fingerprinting using the µXRF scanner. It was primarily intended to ascertain the chemical signature of the archive half surface (split core) and subsequently, with the existing CT-scanning data.

Lines 161 – 162 comment states: "This really requires a bit more detail to be intelligible or reproducible. Were the histograms of greyscale variability between scans equalized?"

> We intend to clarify this section in the following way:
>
> "*The CT-image slices were normalized to the sum histogram of all slices using the 'enhance contrast' function in Fiji (with 0.2% of pixels allowed to reach saturation). Following initial normalization, the brightness and contrast settings of the slices were uniformly adjusted manually using the brightness/contrast adjustment setting of Fiji (ImageJ; Schindelin et al., 2012) for maximum visibility. The normalized and brightness/contrast adjusted CT stacks of each archive core section half were subsequently processed using the trainable 3D segmentation based on the Waikato*

*Environment for Knowledge Analysis (WEKA) environment (Arganda-Carreras et al., 2017; https://imagej.net/plugins/tws/)."*

Lines 165 – 166 comment states: "Please, try to be concrete. This is a lot of fluff, without getting down the specifics."

We intend to revise in the following way:

"*As implemented in Fiji, the WEKA segmentation plug-in utilizes the open-source WEKA machine learning algorithm and provides options to analyze 2D and 3D image sets through a graphical user interface.*"

Line 167 comment states: "Again, very vague."

We disagree, as this is a generalized statement intending to convey the broad capability of the classification/segmentation tool we used. In short, the need for the study is defined by the researchers working on their material. We describe our needs in detail above.

Line 169 comment states: "And...what might be the definition of this? Again, very vague. this is really a pervasive issue in some sections of this manuscript."

We intend to clarify this section in the following way:

"*Sets can be adjusted and refined depending on the outputs of the machine learning classifier until a satisfactory output is reached, based on visual comparison of the classifier output and original images within the WEKA interface (Figs. 2c, 2d)*"

Line 173 comment states: "You use the word training a lot here, but trained on what? There are no steps for verification, correct?"

This term refers to manually classifying images for WEKA image segmentation to learn and reproduce the classification. The training thus pertains to manually classifying instances of visible anomalous grains in several CT slices. We see no issue here. Evaluation was done by comparing the accuracy of the results with the raw images through manual inspection. We intend to clarify this in the following way:

"*Within our selected set of 10 training images, we defined five classes based on key features and used them for the automatic identification of features in the core images (Fig. 2). These selected features include voids (= air), the core liner, D-tube, sediment, cracks within the sediment, and higher-density grains/components within the sediment. For classification based on our training set, we selected the 'FastRandomForest' classifier. The results for each analyzed core section were quantified based on the sum of pixels assigned to each class using the quantification tools in Fiji, totaling $3.88 \times 10^7$ classified pixels. Subsequent trial applications of the trained WEKA classifier outside the selected target sections for µXRF scanning yielded mixed results (see discussion for details).*"

Lines 176 – 177 comment states: "I suppose this might also partly be because of the (problematic) difference in resolution between both scanning methods, but generally not a very encouraging sign."

> The reviewer may have misunderstood the meaning of the text here. µXRF and CT-scanning data were intentionally treated individually and only later compared based on the k-means clustering of the elemental data. Both detections are correct based on the data methodology applied. The issues lie with the contamination, not the process of classification of either dataset.

Line 179 comment states: "Given the emphasis on training, and ground truthing throughout this manuscript, this really surprised me. One sample? When comparing the merit of a new method like this, i think it is key to compare (ground truth) results against traditional methods. As other CT IRD papers have done so in extremis (see, for example, Cederstrøm et al. 2021), I think this is a missed opportunity."

> We disagree with this statement. Further sampling would have been excessive and wasteful, given the limited availability of sample material. As will be made clear later in the text, the detection of the two pipe flake in our first discrete sample, which matched the µXRF signature, made the issues with contamination evident and provided sufficient clarification as to the nature of the detected µXRF signal based on Fe and Mn anomalies. Consequently, as only instances of pipe flake contamination were detected at the surface of our test core sections, further destructive analyses would not have yielded further meaningful results in terms of IRD distribution and comparison of the available non-destructive data. To clarify, we intend to add a relevant clarification to the text of section 3.3:
>
> "*The picked flakes yielded the same distinctive Fe and Mn nature as the µXRF scans, indicating that all surface detections of anomalous grains on our test samples were indeed of the same nature, and are not IRD grains but rather similar metal flakes. Due to this initial confirmation of the likely anthropogenic nature of all surface detections, further discrete sampling and IRD picking was abandoned to conserve sample material (see discussion).*"

Line 187 comment states: "It is not specified here, which - i think - would be pretty useful for readers, but is this the reason for EDS analysis? To add weight to the less quantitative XRF data?"

> This comment is unclear to us. Quantitative EDS analyses were only performed on individual grains to confirm the origin of the anomalous Fe-Mn signatures in the µXRF data. So it was done as a means of identifying the nature of the detected grains, not to 'add weight', as the reviewer puts it.

Lines 191 – 192 comment states: "I get the calcium, but diatoms are silicious?"

> Indeed. We refer the reviewer to the following sentence, which explains the special considerations that need to be taken into account for (biogenic) Si (and Al, as an element commonly present in the clay mineral component of marine sediments).

Line 193 comment states: "See previous comment – confusing"

> See our reply to the previous comment.

Lines 194 to 195 comment states: "So, here, it would be really helpful if information about the radiation source had been provided - see previous comments in the method section."

Agreed, see above reply to the relevant comments.

Line 210 comment states: "Here, some form of ground truthing would be most welcome."

The ground truth is given by the image, which shows the co-occurrence and visual presence of the grain.

Line 213 comment states: "With core surface, you mean scanning surface, I presume?"

We refer to the split surface of the archive half. This will be clarified.

Line 217 to 218 comment states: "One grain?? And what exactly is confirmed here?"

Yes, as this was the amount of surface detections on the split archive half surface. We have clarified the text accordingly.

Line 220 comment states: "What is called a training set here, is more like a classification scheme, right? As in - it is not trained on (validated by) other lines of independent evidence?"

We refer to the WEKA classification training set of manually classified images; this is the correct terminology.

Lines 231 – 234 comment states: "Above, you report that Fe is the signature element to identify anomalous grains. yet, most of these, do not contain significant amounts of Fe..."

Correct, please see further discussion.

Line 238 comment states: "Your work focuses on 2-D slices, so the 3-D bit does not sit so well here."

The generated 2D slices from a medical CT scanner serve as the basis for any 3D visualization. The 3D WEKA segmentation evaluates all slices of a section as a continuum, and all segmentation results can be displayed in 3D. The choice to display mainly flattened or 2D slices is rooted in the fact that scaled 3D renderings of 1.5 m long archive half-core sections (or any exctracted segementation results) are somewhat unwieldy.

Lines 239 – 244 comment states: "Jeah, think the bias is not just potential. Also htink that the offset between both techniques applied here is a major issue with the study design. I appreciate that it is mentioned up-front, but it really limits what we can take from this, especially in light of other, higher resolution efforts to do the same (use scanning methods to detect IRD - see prior comments)"

While we appreciate this insight, it has no bearing on the efforts to apply these methodologies to IODP archive halves, as these technologies are not easily applied to them. We intend to clarify the introduction in the above-noted manner to provide further explanation.

Lines 250 – 260 comment states: "I think this is indeed an in important point, and some form of validation with DBD measurements would have been helpful, but believe that the essence of this point can be made in far fewer words: please tighten."

Thank you for the suggestion, but upon review, we consider the length necessary to convey the inherent nuances of this matter.

Line 265 comment states: "As said before too, and also in light of the SEM-EDS results, I think the link between the (presumably known) provenance of IRD in these records, and your XRF results should be clarified."

The reviewer appears to misunderstand the nature of this paragraph completely or has not thoroughly read the entire section before commenting. The µXRF results are clarified below. This paragraph presents facts, and below follows the interpretation of these facts. We encourage the reviewer to also read lines 281–293 of the original draft, which will provide a detailed explanation of these questions.

Line 271 comment states: "Given that, at this point, it is very clear that both approaches have a very different resolution, I think that it would be wise to tone this down a bit."

We cannot agree with the pervasive argument that "differing resolution" will hamper our interpretation and even compromise the overall validity of our data, which the reviewer has repeatedly raised throughout their review. Logically, there is no need for analyses to be run on the same (pixel-for-pixel) resolution to allow comparison of results in terms of spatial occurrence.

Lines 272 – 273 comment states: "IRD can consist of a variety of different bedrock types, including carbonate rocks. This greatly over-simplifies that reality."

Correct, however, assuming a carbonate matrix, detecting carbonates becomes difficult; hence, the target was lithic grains as a more easily diagnosable feature. As this evidently was too presumptive, we intend to clarify this section in the following way:

"*Ideally, IRD should reflect a typical lithic or carbonatic composition containing Si, Al, Fe, Ti, Ca, Mg, and similar common elements. Due to the commonly carbonate or biogenic silica-rich sediment encountered at Site 266, both Ca and Si are, however, considered non-diagnostic for IRD identification based on µXRF data.*"

Lines 295 – 296 comment states: "Given the set-up and results here, I think this is phrased far too strongly. Best tone this down, please."

While we don't agree with the reviewer's argument, we agree that overly strong phrasing should be avoided. We intend to revise to "*provide a first-order framework for non-destructive high-resolution imaging (…)*".

Line 299 comment states: "Again, in light of the significant offset in resolution attained by both methods, you might want to tone this down a bit."

We disagree with this assessment as a similar resolution is not required for spatial comparison. See our response to the comment at Line 271.

Lines 300 – 302 comment states: "So...why did you not do this? It seems quite odd to mention here that the work could have been done better, without explaining why this was not done?"

This was mentioned in the spirit of transparency. The M6 Jetstream was provided to us during a machine demonstration by courtesy of Brucker Japan. This study should be interpreted as a proof of concept to demonstrate the feasibility of the technology for IODP legacy material. Therefore, we believe such a statement is not only honest but also required.

Lines 302 – 303 comment states: "Another strong statement that, in my opinion, is not appropriate. It is also not substantiated."

We disagree; this is a statement of theoretical fact, not any interpretation on our side. Additionally, Si and Al are the most critical elements, as carbonates (primarily CaCO3) are generally masked by biogenic calcareous sedimentation in most marine settings. Si remains important despite the possible occurrence of biogenic silica (diatoms and radiolaria), as the combination with Al and other Trace elements (like Fe, Ti, and Mn, for instance) remains diagnostic, unlike CaCO3, where the only diagnostic element will be Ca.

We did not reiterate this in the interest of brevity. However, it seems this left this statement somewhat unclear, hence why we intend to revise this in the following way:

"*This is expected to significantly enhance the signal quality for Si and Al, two of the most critical elements when identifying anomalous grains such as IRD on the archive half surfaces. Through better resolution of Si and Al and minor elements such as Fe, Mg, and Mn, Ca, and K, the distribution of biogenic silica may potentially be differentiated from denser Quartz and compound lithogenic grains composed of minerals such as feldspars, pyroxenes, or amphibolites.*"

Line 304 – 306 comment states: "As with the difference in resolution between CT and XRF, the honesty of mentioning this appreciated. However, given that there are studies out there that can isolate materials (i.e. volcanic ash) based on density differences with the sediment matrix of less than 0.5 g/cm3, this is a red flag. This is a huge difference in density, and should have been picked up by classification."

We see how this may lead to some confusion. We intend to revise in the following way:

"*Once accurately calibrated, this could be done with available CT scanning data and the applied WEKA machine learning tool, although false positives are still likely with smaller grains/flakes as the signal aliasing will continue to reduce the apparent density of small drill pipe material when spread out over multipe pixels based on the Nyquist-Shannon sampling theorem (Takase, 2024). Better resolution and differentiation would only be possible through the application of µCT imaging. However, this would only be possible by using a system that does not require the constant rotation and/or tilting of the core material, which was not available to us.*"

Lines 310 – 312 comment states: "I do not disagree with this statement, but do not think that is the main take-away from the above t.b.h."

We disagree with that assessment, as this is indeed the core of our study. Although, upon rereading our statement, some clarification may be in order. Hence, we intend to revise this section in the following way:

*"Ultimately, our non-destructive multi-proxy approach nevertheless offers new insights into the high-resolution imaging of sedimentary core material and highlights the importance of first-order validation of any high-resolution signal. The detection of drill-pipe contamination through µXRF and EDS analysis in our test material highlights that it remains critical to ascertain the origin and chemical composition of high-density grains in legacy core material before attempting meaningful sedimentological and paleoclimatological interpretation."*

Line 319 comment states: "Again, please tone this down - way to strong, based on what is presented."

We value the reviewer's opinion and intend to revise accordingly.

Line 334 comment states: "Not sure ground-truth is the right word to use here."

In the absence of a viable alternative term that we are aware of, we will not change our wording here.

Lines 336 – 337 comment states: "As mentioned earlier on in this review, I believe earlier other studies have done so, in a way that goes beyond what is presented in this work. This reads as if this work is a first and novel attempt - i do not think that is an adequate reflection of reality."

We intend to revise this sentence accordingly.